# Determination of Stress Intensity Factors under Shock Loading Using a Diffraction-Based Technique

**Matúš Turis [1,\*], Oľga Ivánková [1], Peter Burik [2] and Milan Držík [3]**

1   Department of Structural Mechanics, Faculty of Civil Engineering, Slovak University of Technology, Radlinského 11, 810 05 Bratislava, Slovakia; olga.ivankova@stuba.sk
2   ZP Research and Development Centre, Ltd., Železiarne Podbrezová, a. s., Kolkáreň 35, 976 81 Podbrezová, Slovakia; burik@zelpo.sk
3   International Laser Center, Ilkovičova 2961/3, 841 04 Bratislava, Slovakia; milan.drzik@ilc.sk
*   Correspondence: matus.turis@stuba.sk

**Abstract:** An experimental optical method has been developed for the measurement of opening and sliding notch face movements. The light passing through a thin slit is monitored by a photodiode detector. Two parts of the slit are fixed independently on the notch faces of the simulated crack. Dynamic variations of the notch face movements are recorded as an electric signal by an oscilloscope. The sensitivity of such displacement measurement is comparable with the wavelength of light. Dynamic mixed-mode stress intensity factors under shock loading were evaluated from the data obtained and subsequently compared with a numerical simulation by ANSYS software. As it was approved, the technique has shown sufficient sensitivity, good linearity, and measurement reliability. Due to its non-destructive nature and overall robustness, the arrangement is applicable even for structural component condition determination taking into consideration potentially unknown boundary conditions and the non-linear character of mechanical parameters.

**Keywords:** optical measurement; diffraction-based technique; dynamic stress intensity factor; mixed-mode loading

## 1. Introduction

Dynamic stress intensity factor (DSIF) is a crucial parameter in elastodynamic fracture mechanics. Dynamic loading conditions strongly influence stress waves propagations and their reflections from specimen boundaries, and inertial effects resulting from transient loading conditions, and the rate dependency of a material. The propagating stress waves can interfere and thus generate interference intensification or attenuation. Therefore, unlike the static solution, the DSIF cannot be understood as directly proportional to the instantaneous value of the dynamic force. Cumulating the stress waves can cause a higher peak of the stress intensity factor than the static equivalent of a problem. Also, high strain rates might induce brittle fracture of a material that is to be said ductile in static condition.

Determination of DSIF is described analytically, experimentally and numerically by several authors. Cracked Euler-Bernoulli beam, in one-point bending loading mode, is presented in [1]. Two formulae were used to determine the DSIF, one based on the difference between the displacement in the mid-span and that at the end of the beam, and the other based on the bending moment at the mid-span. A similar problem of estimating DSIF using the Timoshenko beam equation [2] and modified Timoshenko beam equation for deriving mode shape functions and frequency equations of a cracked beam is solved in [3]. In the mentioned above, the crack was considered as a discontinuity in the moment of inertia. In [4], the closed form of DSIF and contact force were derived based on the mode superposition technique. Paper [5] presents both experimental and numerical approaches to determine a dynamic fracture behavior of a one-point bending test, and the experimental measurement is provided using a modified Kolsky bar apparatus.

Measurement of DSIF at fracture initiation time at three-point bend specimen and numerical solution for the same problem can be found in [6]. Described methodology of determining the dynamic fracture toughness using a combination of experiments and numerical approach is given in [7]. The value of DSIF was achieved using a measured applied force with a calculated specimen response to a unit impulse force. The derived formula for DSIF of three-point bend specimen using the vibration analysis method and corresponding dynamic test on modified Hopkinson pressure bar is presented in [8]. The dynamic fracture test using Split-Hopkinson pressure bar and measured mixed-mode crack propagation velocity by synchronized measuring system can be found in [9]. The dynamic initiation stress intensity factor was calculated using the experimental-numerical method, and the crack path was predicted numerically. In [10], the effect of holes on dynamic crack propagation under impact loading was investigated, which demonstrated that the suppressing effect on moving crack increased inversely proportional to the spacing of holes.

There are also a lot of numerical methods for describing the behavior of cracked bodies under impulse load. In [11], using finite element analysis to evaluate reaction force in three-point bend specimens, the effect of time step length, the number of considered eigenmodes, and specimen overhang on the achieved anvil force and DSIF were investigated. The meshless finite block method used for the calculation of mixed-mode dynamic stress intensity factors is introduced in [12]. Another meshless material point method and a description of algorithms for fracture parameters in 3D dynamic problems and their comparison with finite element method (FEM), finite difference method, dynamic boundary element method, and static theories are listed in [13].

This paper presents an experimental-numerical study of the possibility of applying an optical technique that can be used for the experimental evaluation of DSIF. The basic principle of the method consists of the photoelectric sensing of light passing through a slit mounted on notches simulating a crack. With this approach, it is possible to measure fast notch face dynamic movements in mutually perpendicular directions separately. When applying a relatively short force impulse due to a specimen's self-period of oscillation, it can be used as a non-destructive method for describing the structural component condition. On the other hand, it can be used for non-propagating both slow and fast crack face movements and static conditions.

## 2. Materials and Methods

### 2.1. Diffraction-Based Monitoring of Notch Face Movements

When light penetrates the narrow slit, the light intensity at the far-field behind it is described by a well-known Fraunhofer diffraction pattern. We assume the illumination to be a normal incident plane-wave field of amplitude U(x). We also assume $z \gg w$, where z and w are the distance slit—observation plane and width of the slit, respectively. The transmittance function—the diffraction integral gives the complex amplitude at the observation plane $(x_0, y_0)$:

$$U(x) = U(x) w \mathrm{sinc}\left(\frac{wx_0}{\lambda z}\right), \tag{1}$$

where $\lambda$ is the wavelength of light and the function:

$$\mathrm{sin\,c}\left(\frac{wx_0}{\lambda z}\right) = \sin\left(\frac{\pi wx_0}{\lambda z}\right) / \left(\frac{\pi wx_0}{\lambda z}\right). \tag{2}$$

Thus, the intensity distribution registered by the quadratic detector is:

$$I(x_0) = |U(x)|^2 \left(\frac{w}{\lambda z}\right)^2 \mathrm{sin\,c}^2\left(\frac{wx_0}{\lambda z}\right) \tag{3}$$

This is a well-known form of irradiance with the expressive central peak of maximum intensity and small alternating maxima and minima, which vanishes on both sides of the pattern. As seen from Equation (3), the light behind the slit varies as a square of slit width.

If an effective area of a photoelectric sensor with linear characteristics and quasi point-like dimensions is positioned to the maximum intensity, the output electric signal depends quadratically on the slit width too. The system's response to the slit opening (or closing) is non-linear, and correction will be needed in this case.

However, as the amount of energy in the central part of the diffraction pattern is dominant, this effect affects the accuracy a little. Besides that, even though the relative change of slit opening under the stress waves deformations is very small, a reliable way to make the output linear is to use an integral intensity collection by a wide-angle lens collimator. In this way, the image of the slit is projected on to a photodiode effective area. In such an arrangement, the photodiode is illuminated by nearly all the light passing through the slit.

The more significant problem of such optical tracking is to achieve the system to be invariant with regard to undesirable motions of the measured slit relative to the laser beam trace—rigid body motion of the object under loading. This is very important in dynamic conditions considering outside vibrational influences and possible motions of the whole measured object. We can neglect small rotational movements around both the axes—rotational of the laser beam and parallel to the slit edges. In the latter case, an apparent width of the slit varies as a cosine factor of the rotational angle; consequently, the light irradiance behind the slit does not change more than ~1%. Much more dangerous is a shift of the slit perpendicularly to its edges outside the laser beam center. Usually, CW laser resonators emit $TEM_{00}$ mode of radiation with Gaussian transverse irradiance profile. Any displacement of the narrow slit in a perpendicular direction will give a considerable false output signal. Therefore, an essential condition is to create a transversely early constant intensity distribution at the plane of the slit. It can be realized using a telescopic optical system or simply by sufficient enlargement of the laser beam in this direction.

The arrangement shown in Figure 1 has been adjusted in which a laser beam forms the measurement basis, and a photodiode detector is a receiver of the optical signal. In experiments, the laser diode module with wavelength $\lambda = 640$ nm, output power 24 mW, and elliptical cross-section $TEM_{00}$ mode with dimensions of ~ 6 mm × 2 mm, was used. The optical signal was sensed by a photodiode Thorlabs 3.6 mm × 3.6 mm Si biased detector DET 36A. The effective detector area is optimal for laser spot reliable adjustment and a response of 14 ns rise time is also reasonable to monitor dynamics of mechanical stress waves. The detector offers the possibility to amplify the signal up to 70 dB.

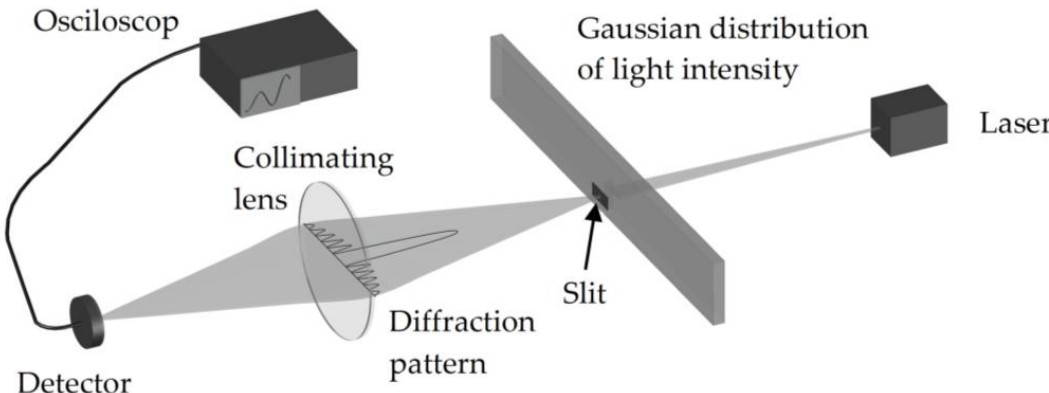

**Figure 1.** Scheme of the experimental setup for monitoring notch face movements.

Since the illumination intensity of the detector generates the static signal, which significantly exceeds its dynamic variations in the measurement, the modulation of the signal is relatively small. For compensation of the static signal component, it is necessary to connect the detector either in the mode of the electric bridge or use the AC mode of oscilloscopic recording or, alternatively, its offset compensation.

On the model, an aperture of the slit was formed by two sharp edges of thin steel plates/razor blades. Those were mounted in a point-like manner, using adhesive, onto a steel specimen on both sides of the crack simulated notch. As the principle of superposition can be used to deal with stresses around the crack tip, the results of the general case solution can be obtained by superposing three basic loading modes. Particular modes differ from each other in the orientation of the external load relative to the notch plane and in the direction of relative movement of the two notch faces. When antiparallel movements are to be detected (shearing mode), the orientation of the slit is perpendicular to the notch with the fixation of the measuring blades on opposite sides of the cut. As a result, the two mutually perpendicular or antiparallel dynamic movements can be evaluated.

The width of the slit was precisely prearranged by means of a diffraction pattern projected onto the testing screen. The width of the slit was chosen so that the modulation of intensity due to stress waves was sufficient with minimal unwanted noise. The optimum width of the slit can be adjusted to be as narrow as possible, the diffraction pattern of which is reliably collected by a collimator lens. As a collimator, a photographic objective with max aperture number 1:1.4 was used.

### 2.2. Measurement and Results Evaluation

To test the proposed measurement technique, an experimental study of two variants of notched steel specimens was carried out. Both the beam and L-shaped specimens (Figure 2) were positioned on rubber anvils supports. The second L-shaped specimen also had stabilization supports, realized by rubber from three sides to prevent contact with a fixed clamping mechanism. The clamping screws have been tightened only to the extent necessary to freely hold the specimen in place with no additional pressure. As known, the comparison of numerical simulation results with the experiment is often affected by the non-ideal rendering of the boundary conditions of the structural element fixation. Thus, when the experimental and numerical results are to be compared, the boundary conditions play a significant role. Especially in dynamic loading conditions, such factors as support friction, loss of contact with supporting element, stiffness of contact materials and anvils, or vibration of the supporting structure cannot be neglected.

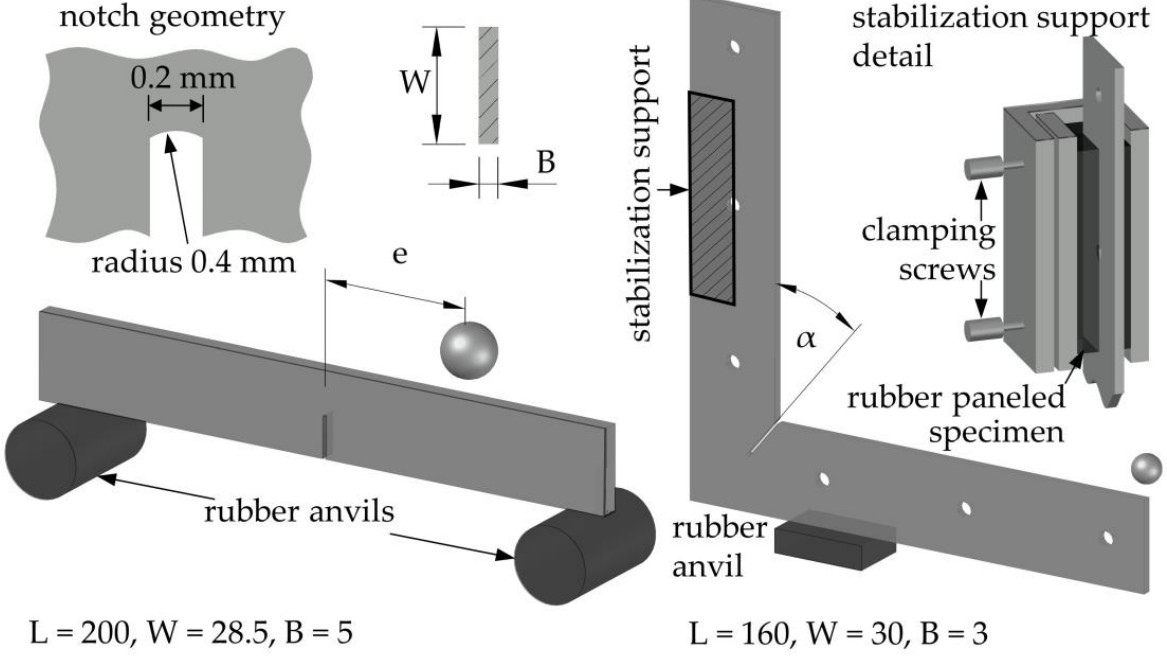

**Figure 2.** Geometry and boundary conditions of the investigated bodies (mm). L is the length of the given specimen.

To prevent this effect of specimens mounting, soft supports with orders of magnitude lower natural frequency than the specimen frequencies were chosen. These frequencies were identified in spectral analysis with values of 83 Hz and 11 Hz for the beam and L-shaped specimen measurement, respectively. In such a case, the specimens can be considered as freely supported in a numerical model. No additional algorithms were then needed to find the contact between the specimen and supporting structure, and experimental measurements validation can thus be considered more reliable.

Figure 2 shows the specimen's geometry. Both the elements were subjected to different loading values and load point positioning. The dynamic force was excited by the free fall of steel balls having 17.5 mm diameter for the beam specimen and 9.5 mm for the L-shaped specimen. The path of a free fall was, in both cases, one meter. The beam specimen was loaded in both symmetric pure mode I loading (one-point bending) and mixed-mode I/II loading conditions. To excite mixed mode for beam specimen, a non-symmetrical positioned load point was used. The Eccentricity of the impact point was then e = 0 mm and 30 mm for pure mode I and mixed-mode loading, respectively. For the L-shaped specimen, the angle orientation of the notch was chosen $\alpha = 45°$, and a/W ratio was 0.5 in both cases, where a was the crack length.

Using tensile testing, the Young's modulus of elasticity E = 202.44 GPa and 207.43 GPa, Poissons ratios $\nu = 0.29$ and 0.30 were achieved, and densities $\rho = 7857.78$ kg·m$^{-3}$ and 7738.47 kg·m$^{-3}$ were determined pycnometrically for beam and L-shaped specimens.

To demonstrate the influence of stress waves with their free boundary reflections, the time duration of applied impulse force should be chosen to be several times shorter in comparison with the specimen's natural vibration period. Figure 3 shows the frequency content of in-plane vibrations of both the specimen examples obtained by signal analysis of the recorded notch face movements. Notation opening and sliding in Figure 3 means that the frequency spectrum was achieved from a mode I and mode II measurement setup, respectively.

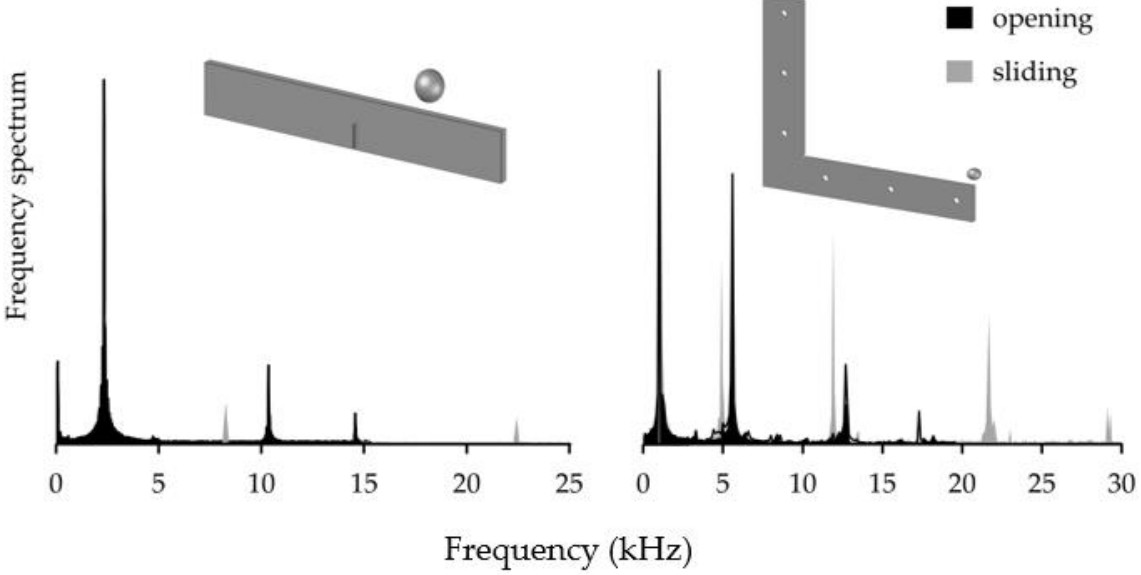

**Figure 3.** Frequency spectrum for beam and L-shaped specimen.

For a more complex evaluation of the data obtained using the diffraction method, absolute values of the measured quantities need to be known. Therefore, the time course and the calibration of the impulse force values induced by the free-falling ball were determined using a piezoceramic sensor. Recorded shock force waveforms, when hitting both the larger and smaller balls on the sample, are plotted in Figure 4. It shows the signals of the piezoceramics slab fixed between the sample and the striking ball. From this, the shape

of force-time function and impulse duration time was known. By comparing these results with the theory of elastic impact, the impulse force-time record was achieved. As it can be seen, the shape of the impulse force meets the characteristic half-sine course of time-force function predicted by Hertz's theory of elastic impact [14,15].

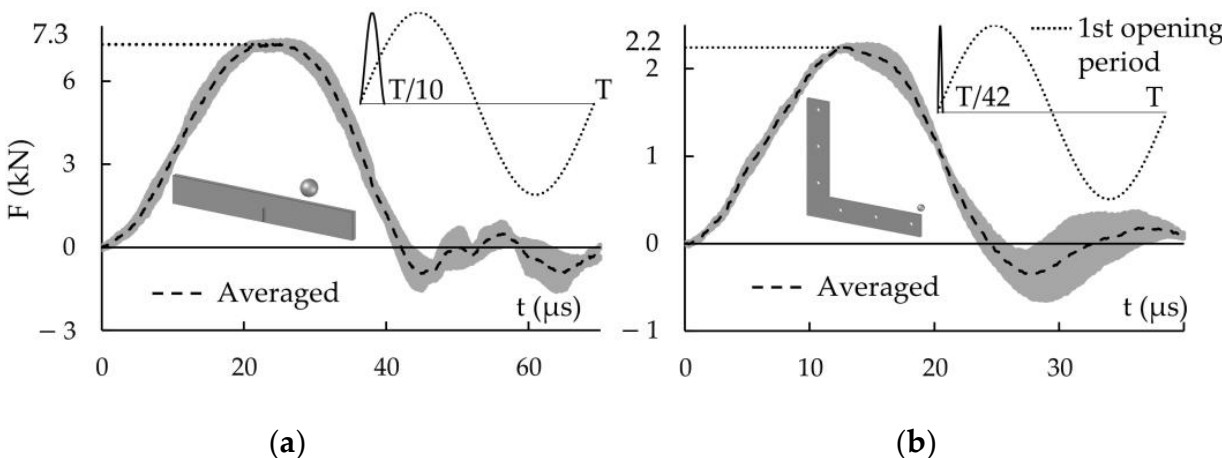

**Figure 4.** Measured impulse forces and their comparison to the first notch opening displacement period: (**a**) beam specimen D = 17.5mm; (**b**) angled specimen D = 9.5 mm where D is a diameter of a steel ball. The grey color represents the experiment range.

In the numerical simulation, only the first positive half-sine wave was considered. In Figure 4, loading force peaks are also plotted when they are confronted with the period of natural frequency vibrations.

*2.3. Dynamic Stress Intensity Factors Evaluation-Experiment vs. Finite Element Method (FEM)*

When the task to assess the time development of stress intensity factor and its peak value determination under impact loading of the structure is assigned, the value of measured DSIF can be computed directly from the asymptotic displacement field around the crack tip [16]:

$$K_i(t) = \lim_{r \to 0} \frac{E\Delta_{ax}(\eta_r, t)}{8(1 - \nu^2)} \sqrt{\frac{2\pi}{r}} \tag{4}$$

where

$i$ considered mode I, II,
$K_i(t)$ the stress intensity factor of $i$th mode in time $t$,
$\Delta_{ax}$ time-dependent displacements difference in the local coordinate system; ax = x for $K_{II}$, y for $K_I$,
$\eta_r$ point on the notch face in distance $r$ from notch tip,
E Young modulus of elasticity,
$\nu$ Poisson ratio,
r distance from notch tip.

Nevertheless, the Equation (4) presents only the first term in the displacement field series of asymptotic analysis around the crack tip. Since the notch face movements were measured only in one point on notch faces, direct evaluation of the stress intensity factor would be affected by an error. In experimental measurements, the laser beam illuminated the prearranged slit in a chosen position with polar coordinate r = 6 mm from a notch tip. When using only the first term of analytical displacement field solution, an obtained error calculating the DSIF directly from notch face movements in the located region was around 30%. For a more accurate evaluation of the stress intensity factor from the experimental data, it was needed to measure the movements of the set of points lying on the traction-free

crack surface. This leads to several lasers and detectors that are recording notch face movements simultaneously. Due to this, it is practical to assign the stress intensity factor to measured notch face movements numerically.

Numerical modeling and the calculation of the dynamic stress intensity factors have been performed by a finite element method using ANSYS software (Canonsburg, PA, USA). A 2D finite element with quadratic displacement behavior and two degrees of freedom for each node was selected. The chosen type of element consisted of eight nodes in a rectangular shape and could be degenerated into the six-nodes triangular shape element. The task was solved as a plane strain state problem. A linearly isotropic material model was considered. For modal analysis, the block Lanczos mode extraction method was used and, for transient analysis, the Distributed sparse matrix direct solver was applied.

Using numerical analysis, the effect of the notch geometry as the simulated crack has been evaluated. The geometry of the notch is shown in Figure 2, and a sharp crack is shown in Figure 5. It has been found that this has a minor effect on the observed results of notch face movements in a given position from a notch/crack tip ±3.27% between obtained local extremes of the time-dependent notch and crack face movements function. Therefore, numerical results of the real notch geometry from Figure 2 were used for notch face movements comparison with experimental results, and crack-like geometry was used to assign the DSIF values to achieved notch face movements.

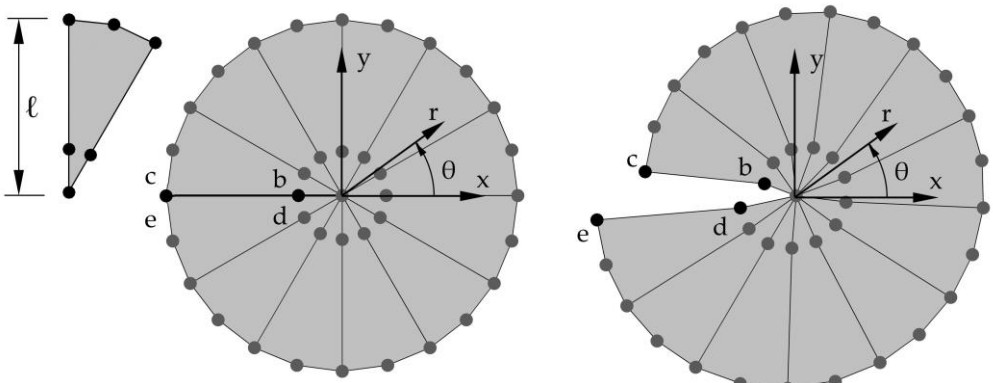

**Figure 5.** Singular element, polar coordinate system and nodes needed for node displacement extrapolation method.

The numerical model of the beam specimen consisted of 6227 elements/19,188 nodes, and 6589 elements/20,254 nodes for notch and crack-like geometry was used, respectively. The L-shaped specimen consisted of 9632 elements/29,721 nodes, and 10,456 elements/32,197 nodes for notch and crack-like geometry, respectively. Global finite element edge length in both cases of notch/crack geometry was set as one millimeter to capture the propagation of stress waves in the material. In the notch/crack vicinity, the mesh was refined with respect to isolines of the equivalent Von-Mises stress.

Finite elements for the purpose of DSIF determination were arranged rotationally around the crack tip. In the first row, the elements were collapsed to the triangle shape and the center nodes were shifted to a quarter of the element's length, Figure 5. Such elements are suitable for capturing the singularity of stresses in the crack tip.

For numerical estimation of DSIFs, the node displacement extrapolation method (NDEM) was used [17].

When considering only mode I and II and plane strain conditions, the following equation can be written:

$$K_i(t) = E \frac{8\Delta_{ax1}(t) - \Delta_{ax2}(t)}{24(1 - \nu^2)} \sqrt{\frac{2\pi}{\ell}}$$

(5)

$$\Delta ax_1(t) = u_{axb}(t) - u_{axd}(t) \quad \Delta ax_2(t) = u_{axc}(t) - u_{axe}(t)$$

where

*i* considered mode I, II,

$u_{axm}(t)$ time-dependent displacements in the local coordinate system; ax = x for $K_{II}$, y for $K_I$; m = b, d, c, e,

$\Delta_{axj}(t)$ time-dependent displacements difference in the local coordinate system; *j* = 1, 2,

$\ell$ length of the element.

Preliminary finite element analysis results were also used for the experimental setup alignment. From known material properties and specimen geometry, first mode shapes were calculated. Based on obtained eigenfrequencies, length and number of time steps were selected for the purpose of supposed transient analysis. The finite element model was continuously updated in the sense of mesh density adjustment, loading with measured force, and time step control settings.

### 3. Results

In this section, experimental results are listed and compared with those obtained by finite element analysis. The results of measured two perpendicular notch face movements are introduced.

The main experimental results obtained are two mutually perpendicular movements of notch faces. The recorded light intensity changes related to notch opening displacements are shown in Figure 6. The boundary condition of a task can be specified as a one-point bending, where the impulse force is applied to a mid-span and the whole problem is symmetric.

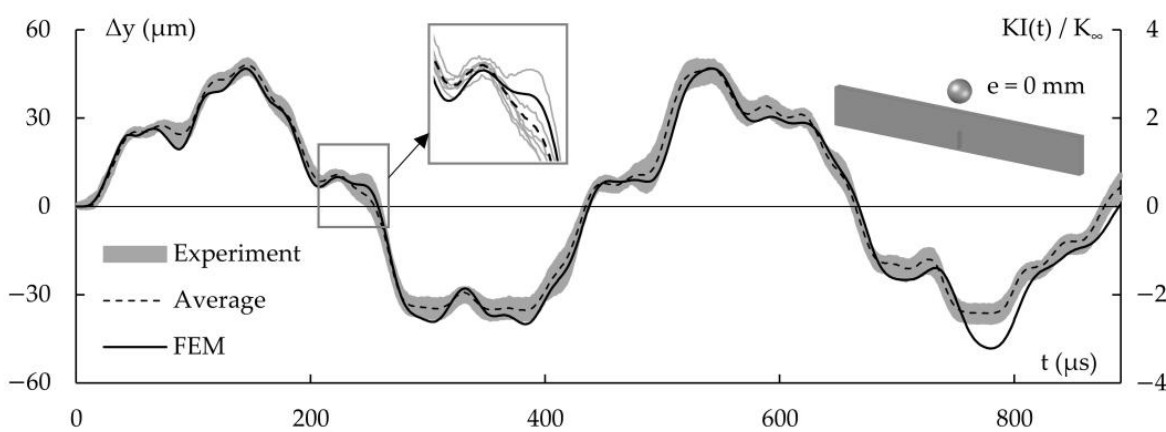

**Figure 6.** Time dependences of notch opening displacements obtained by diffraction-based measurements and the curve of FEM calculation. The grey color represents the experiment range.

In Figure 6, several experimental measurements are shown covered by a grey area indicating the variance of experimental values. The notch face movements were recorded at a distance r = 6 mm from notch tip and angle orientation θ = 180° at free notch boundaries.

Using a calibrated slit with a micrometer screw, the calibration curve was plotted as a photoelectric signal data vs. adjustable slit width. After matching a slit opening that gives the same light intensity as a specimen slit in rest, the adjustable width was opened and closed by a small amount and light intensity was recorded. This way, the output detector signal was assigned to an adjustable slit opening in meters and achieved the ratio of volts to meters.

The time development of mixed-mode dynamic stress intensity factors for beam specimen with loading point eccentricity e = 30 mm, are displayed in Figure 7. Plotted DSIF values are normalized with $K_\infty = \sigma(\pi a)^{0.5}$, where $\sigma = F_{max}/BW$.

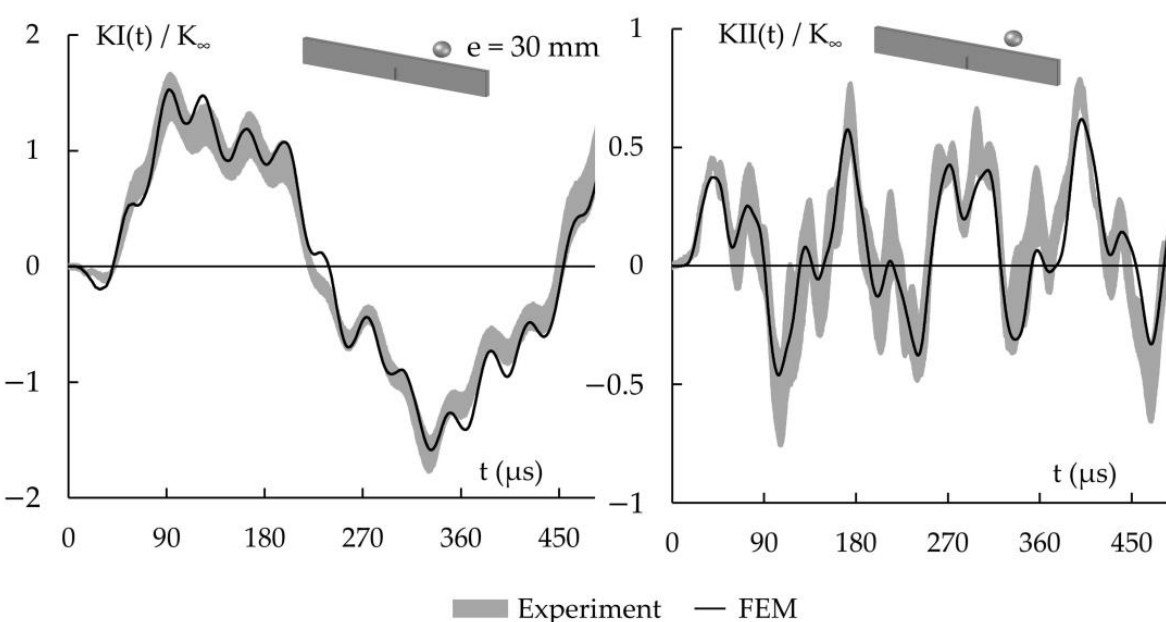

**Figure 7.** Measured and computed DSIF for beam specimen loaded with e = 30 mm from midspan. The grey color represents the experiment range.

An L-shaped specimen with notch orientation $\alpha = 45°$ is shown in Figure 8. It is another illustrative example where both in-plane DSIFs are evaluated.

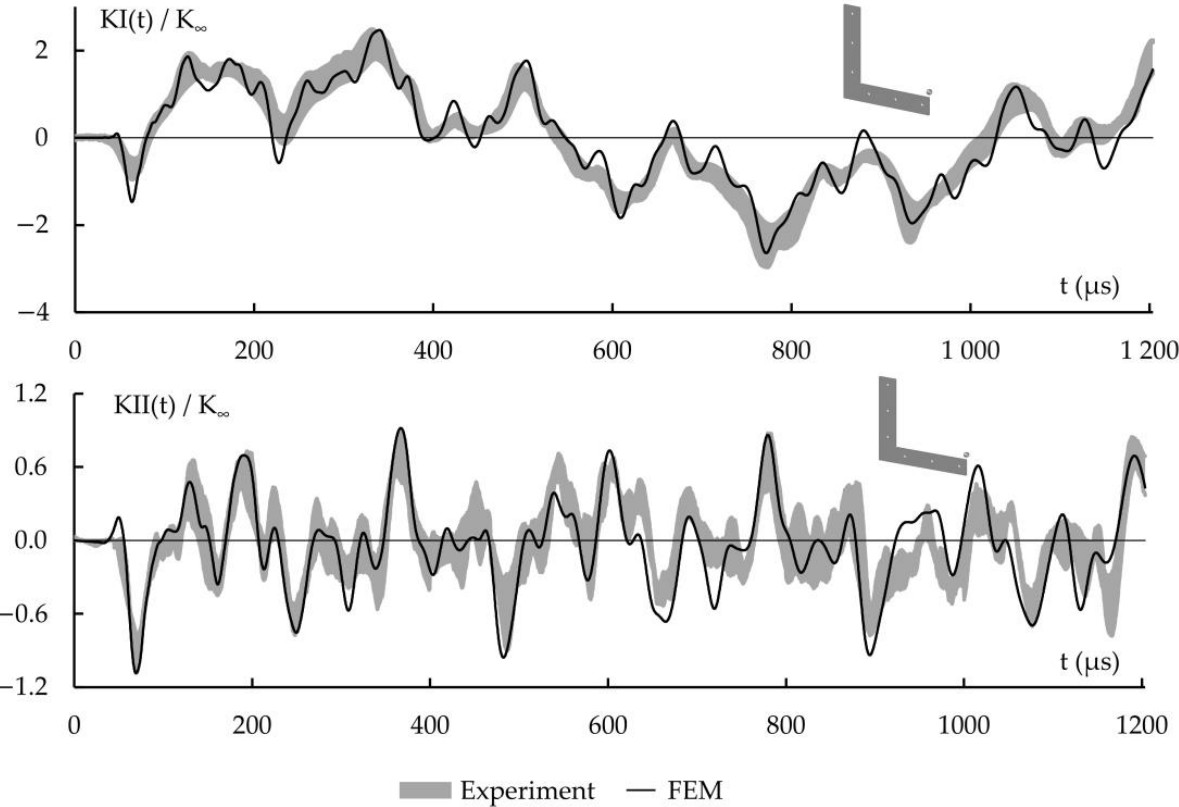

**Figure 8.** Measured and computed DSIF for angled specimen with notch orientation $\alpha = 45°$. The grey color represents the experiment range.

The duration of time step in numerical analysis was set with respect to the highest measured frequency. To achieve a correlation of results as presented above, approximately 150 steps for the lowest measured period were needed. That means 2430 and 4800 time-steps for beam and L-shaped specimen, respectively. In measured time records of mode II, mode I frequency was also identified, and the sliding frequency was oscillating around it. In the graphical presentation of mode II, shown in Figures 7 and 8, this frequency was subtracted so only the sliding mode was presented.

Figure 9 shows the graphical evaluation of the percentage error of achieved results. This figure expresses the percentage of the individual measured values of time-dependent notch face movements that meet the given error range between experimental and numerical results. In this comparison, the results obtained numerically were determined as reference values.

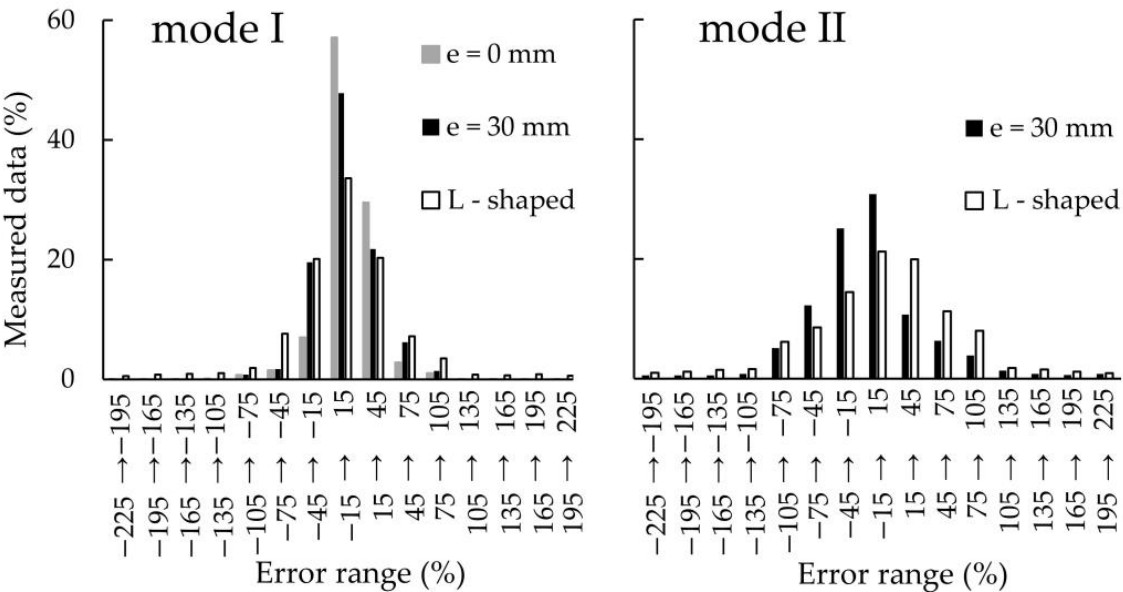

**Figure 9.** Percentages of measured data that satisfy the error range.

## 4. Discussion

As seen from presented examples of dynamic deformation time developments, the implemented method using light diffraction to determine the crack (or notch) opening meets the measuring method's requirements, enabling the analysis of the dynamic phenomena of the cracked body. The results of notch face movements presented in Figures 6–8 in the form of the dynamic stress intensity factor, obtained from such experimental diffraction-based techniques, show a close similarity with those from finite element analysis. Precise prearranged slits provides high sensitivity to light intensity changes, i.e., notch face movements. Small differences obtained in individual measurements can be caused by slightly uneven loading. Using a piezoceramic sensor, the variational maximum error of impulse value $\int Fdt$ less than 2% was obtained. Another possible reason for differences can be specimen position change after individual measurement due to a rigid body motion that can cause slightly alternating impact point position. The repeatability of the impact point was ensured by falling through the pipe with a half-millimeter larger diameter than a ball. The maximum distance between the two impact points was identified as 1.7 mm. Numerical analysis results show differences in monitored results between these two boundary points, ±0.85 mm, from desired impulse position. The mean absolute percentage error of considered time steps between these boundary points was 4.46–4.78% and 4.57–4.67% for mode I and mode II, respectively. These effects can be suppressed by a reasonable number of measurements concerning specimen surface abrasion. From the statistical evaluation of the results in Figure 9, a typical Gaussian distribution can be seen, with a maximum for the

range of ±15% between the results obtained experimentally and numerically. It can also be noted that the measured results reflect very well the trend of time-dependent notch face movements function achieved numerically in terms of shape and amplitudes.

The first attempts in experimental realization showed significance to ensure the boundary conditions similar to numerical assignment as well as possible. In particular, the fixation of the structural elements proved to be most important when the results are confronted to one another. Contact between specimen and mounting can be difficult to specify precisely in numerical simulation, especially in dynamic conditions where the transfer of a supporting frame's own vibrations is challenging to isolate. Therefore, anvils that freely hold specimens in position with no additional pressure have been used. By applying an anvils material with orders of magnitude with lower eigenfrequencies, this effect can be almost eliminated. Due to that mentioned, the experimental and numerical approach results served in good agreement.

## 5. Conclusions

The developed measuring method represents a relatively simple optical setup, which can be implemented in a laboratory as well as in non-laboratory conditions. As evidenced by comparing experimentally and numerically obtained results, the proposed measurement method and methodology for evaluating the results are entirely acceptable in experimental measurement. This is shown by graphical dependences of some selected time-dependent notch face movements from several performed measurements. Due to the factors influencing the experimental measurements, the errors obtained can be considered satisfactory. Nevertheless, thanks to sufficient sensitivity of the measurement, the possibility of monitoring even very fast mechanical events, and its robustness, the method can be applied in many applications of fracture mechanics. Based on achieved results and approved good coincidences with numerical analysis results, its reliability and accuracy can be pointed out. The method can be potentially used in a real environment investigation in non-linear fracture mechanics and when the boundary conditions, especially in complicated cracked constructions, are difficult to define. Then even evolution algorithms or another suitable procedure can be used to obtain the correct boundary condition for the more profound observation of the cracked body in numerical simulation.

**Author Contributions:** Conceptualization, O.I. and M.D.; methodology, M.T., O.I. and M.D.; software, M.T.; validation, M.T. and M.D.; formal analysis, M.T., O.I. and M.D.; investigation, M.T., P.B. and M.D.; resources, P.B. and M.D.; data curation, M.T. and M.D.; writing—original draft preparation, M.T. and M.D.; writing—review and editing, O.I. and M.D.; visualization, M.T.; supervision, O.I., P.B. and M.D.; project administration, O.I. and M.D.; funding acquisition, O.I. All authors have read and agreed to the published version of the manuscript.

**Funding:** This research was funded by the Grant Agency of the Ministry of Education, Science, Research and Sports of the Slovak Republic SK-KEGA 025 STU-4/2019.

**Conflicts of Interest:** The authors declare no conflict of interest.

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
