# Peer review of "Determination of Stress Intensity Factors under Shock Loading Using a Diffraction-Based Technique"

_applsci, doi:10.3390/app11104574_

Round 1

Reviewer 1 Report

Please, find the attached document including comments to your article.

Author Response

I would like to thank you for your revision.

Following changes were made in the manuscript as you requested.

Point 1:  Lines 36, 37: there is syntactic mistake. This sentence should be changed.

Response: Syntactic mistake was fixed as follows

Determination of DSIF is described analytically, experimentally, and numerically by several authors. 

lines 36-37 in the uploaded document

Point 2: Lines 57: there is no explanation of the “SIF” abbreviation.

Response: The abbreviation has been substituted with the whole meaning

lines 57 in the uploaded document

Point 3:  Line 69: there is no explanation of the “FEM” abbreviation.

Response: The FEM term has been given its full name, the abbreviation has been given in parentheses for repetition in the text

line 69 in the uploaded document

Point 4: Lines 84: the sign “>>” is to small . 

Response:  The designation has been replaced by a larger variant

line 84 in the uploaded document

Point 5: Equations 1-3: there are no explanation of several symbols like x0, z 

Response: The x0 was described as a point in the observation plane (x0, y0) (line 87 in the uploaded document). Lambda is a wavelength of light (line 88 in the uploaded document). The z is  the distance slit – observation plane (line 85 in the uploaded document)

Point 6:  Line 122:there is no unit after 3,6 number.

Response: There is added unit (mm)

line 122 in uploaded document

Point 7:  Line 173: there is syntactic mistake. It should be “in” before Figure 3.

Response: The "on" was replaced with "in"

line 188 in the uploaded document

Reviewer 2 Report

The manuscript “Determination of stress intensity factors under shock loading using a diffraction-based technique” describes an optical method for the measurement of notch face movements.

Section 2.2 describes the measurements without taking the assumed values. The content is general, e.g., “low stiffness”, “lower natural frequency”,  “different loading values”, “small steel balls”. The authors concluded “…the boundary conditions play a significant role.” (page 4, line 155) As an important factor, why are the FEM boundary conditions not shown as a diagram? The manuscript does not provide details of the notch geometry. Does geometry affect the result? Section 2.2 should be rewritten.

Section 2.3. What determines r (6 mm)? The details of the numerical model are missing (degrees of freedom of a finite element, number of nodes/elements, material model, discretization error). The statement “It is a simple, precise, and computational time-saving method for cracks as presented in this paper.” (page 6, line 216-217) is general.

Section 3, Fig. 6 contains results for conditions other than those included in the previous sections (Fig. 2). The structure of the manuscript is confusing.

The measurement technique should have a calculated error. If the error depends on the input quantities, the parameters should be explained. The discussion should include an analysis of research results and measurable quantities.

The conclusions are general. The conclusions should be proposed based on the quantitative method for experimental data and numerical calculations.

Author Response

I would like to thank you for your response.

The following changes have been made in the manuscript, as you requested.

Point1: Section 2.2 describes the measurements without taking the assumed values. The content is general, e.g., “low stiffness”, “lower natural frequency”,  “different loading values”, “small steel balls”. The authors concluded “…the boundary conditions play a significant role.” (page 4, line 155) As an important factor, why are the FEM boundary conditions not shown as a diagram? The manuscript does not provide details of the notch geometry. Does geometry affect the result? Section 2.2 should be rewritten.

Response 1:

a) Frequency of rubber anvils has been described based on the frequency spectra of measurement as follows:

To prevent this effect of specimens mounting, soft supports with orders of magnitude lower natural frequency were chosen. These frequencies were identified in spectral analysis with values of 83 Hz and 11 Hz for beam and L-shaped specimen measurement, respectively. (lines 162 - 165 in the attached document)

b) The term "different loading values" has been emphasized  by reference to Figure 4 for beam and L-shaped specimen. (lines 169-170 in the attached document)

c) The term "small steel balls" has been replaced by "steel balls" and its specific diameters were named. (lines 170 - 172 in the attached document)

d) The boundary conditions of the task has been shown in replaced Figure 2 and described as follows:

Both, the beam and L-shaped specimens (Figure 2) were positioned on rubber anvils supports. The second L-shaped specimen also had stabilization supports, realized by rubber from three sides to prevent contact with a fixed clamping mechanism. The clamping screws have been tightened only to the extent necessary to freely hold the specimen in place with no additional pressure.  (lines 150 - 155 in the attached document)

e) the notch geometry has been added to replaced Figure 2. The effect of notch geometry on measured quantities was described as follows:

"Using numerical analysis, the effect of the notch geometry as the simulated crack has been evaluated. The geometry of the notch is shown in Figure 2, and a sharp crack is shown in Figure 5. It has been found that this has a minor effect on the observed results of notch face movements in a given position from a notch/crack tip ± 3.27 % between obtained local extremes of time-dependent notch and crack face movements function. " (lines 236 - 240 in the attached document)

Point 2: Section 2.3. What determines r (6 mm)? The details of the numerical model are missing (degrees of freedom of a finite element, number of nodes/elements, material model, discretization error). The statement “It is a simple, precise, and computational time-saving method for cracks as presented in this paper.” (page 6, line 216-217) is general.

Response 2:

a) The r = 6 mm was chosen distance from a notch tip that was confronted with numerical simulation. It was pointed out as follows:

In experimental measurements, the laser beam illuminated prearranged slit in a chosen position with polar coordinate r = 6 mm from a notch tip.

(lines 218 - 220 in the attached document).

b) Details of numerical model has been added :

A 2D finite element with quadratic displacement behavior and two degrees of freedom for each node was selected. The chosen type of element consisted of eight nodes in a rectangular shape and could be degenerated into the six-nodes triangular shape element.  (lines 229 - 232 in the attached document)

The numerical model of beam specimen consisted of 6227 elements/ 19188 nodes, and 6589 elements/ 20254 nodes for notch and crack-like geometry was used, respectively. The L-shaped specimen consisted of 9632 elements/ 29721 nodes, and 10456 elements/ 32197 nodes for notch and crack-like geometry, respectively. Global finite element edge length in both cases of notch/crack geometry was set as one millimeter to capture the propagation of stress waves in the material. In the notch/crack vicinity, the mesh was refined with respect to isolines of the equivalent Von-Mises stress. (lines 244 - 250 in the attached document)

Linearly isotropic material model was considered.  (lines 233 in the attached document) & Using tensile testing the Young’s modulus of elasticity E = 202.44 GPa and 207.43 GPa, Poissons ratios ν = 0.29 and 0.30 were achieved, and densities ρ = 7857.78 kg·m-3 and 7738.47 kg·m-3 were determined pycnometrically for beam and L-shaped specimen. (lines 179 - 181 in the attached document)

c) The statement "It is a simple, precise, and computational time-saving method for cracks as presented in this paper." was deleted (lines 256 - 257 in the attached document)

Point 3: Section 3, Fig. 6 contains results for conditions other than those included in the previous sections (Fig. 2). The structure of the manuscript is confusing.

Response 3:

Figure 6 and Figure 2 have been changed. The loading condition for beam specimen has been more detailed described in Section 2.2 as follows:

Beam specimen was loaded in both symmetric pure mode I loading (one-point bending) and mixed mode I/II loading conditions. To excite mixed mode for beam specimen, a non-symmetrical positioned load point was used. The eccentricity of the impact point was then e = 0 mm and 30 mm for pure mode I and mixed-mode loading, respectively. (lines 173 - 177 in the attached document)

Axis in Figure 6 has been changed  from previously Δy(V) to Δy (µm) and previously Δy (µm) to KI (t) / K∞

Point 4: The measurement technique should have a calculated error. If the error depends on the input quantities, the parameters should be explained. The discussion should include an analysis of research results and measurable quantities.

Response 4:

a) The calculated error for the measurement technique has been added as a graphical evaluation in Figure 9.

The description of percentage error obtaining is as follows:

Figure 9 shows the graphical evaluation of the percentage error of achieved results. This figure expresses the percentage of the individual measured values of time-dependent notch face movement that meet the given error range between experimental and numerical results. In this comparison, the results obtained numerically were determined as reference values. (lines 304 - 308 in the attached document).

&

From the statistical evaluation of the results in Figure 9, a typical Gaussian distribution can be seen, with a maximum for the range of ±15% between the results obtained experimentally and numerically. It can also be noted that the measured results reflect very well the trend of time-dependent notch face movements function achieved numerically in terms of shape and amplitudes. (lines 336 - 340 in the attached document)

b) Dependency of error due to input quantities has been described as follows:

Small differences obtained in individual measurements can be caused by slightly uneven loading. Using a piezoceramic sensor, the variational maximum error of impulse value ∫Fdt less than 2% was obtained. Another possible reason for differences can be specimen position change after individual measurement due to a rigid body motion that can cause slightly alternating impact point position. The repeatability of the impact point was ensured by falling through the pipe with a half-millimeter larger diameter than a ball. The maximum distance between two impact points was identified as 1.7 mm. Numerical analysis results show differences in monitored results between these two boundary points, ± 0.85 mm, from desired impulse position. The mean absolute percentage error of considered time steps between these boundary points was 4.46% - 4.78% and 4.57% - 4.67% for mode I and mode II, respectively. (lines 321 - 335 in the attached file)

Point 5: The conclusions are general. The conclusions should be proposed based on the quantitative method for experimental data and numerical calculations.

Response 5: To the conclusion has been added following sentences:

As evidenced by comparing experimentally and numerically obtained results, the proposed measurement method and methodology for evaluating the results are entirely acceptable in experimental measurement. This is shown by graphical dependences of some selected time-dependent notch face movements from several performed measurements. Due to the factors influencing the experimental measurements, the errors obtained can be considered satisfactory.  (lines 353 - 359 in the attached files)

I believe that thanks to the incorporation of your comments, the quality of the contribution has increased, and making it a more comprehensible and better process. I want to thank you again for your review.

Round 2

Reviewer 2 Report

The revised manuscript takes into account the comments presented in the review. The manuscript can be accepted for publication.